# Analysis of Styrene-Butadiene Based Thermoplastic Magnetorheological Elastomers with Surface-Treated Iron Particles

**DOI:** 10.3390/polym13101597

**Published:** 2021-05-15

**Authors:** Arturo Tagliabue, Fernando Eblagon, Frank Clemens

**Affiliations:** Laboratory for High Performance Ceramics, Empa, Swiss Federal Laboratories for Materials Science and Technology, Überlandstrasse 129, 8600 Dübendorf, Switzerland; arturo.tagliabue@empa.ch (A.T.); fernandoeblagon@lankhorsteuronete.com (F.E.)

**Keywords:** magnetorheological elastomer, thermoplastic elastomer, magnetorheological effect, static and dynamic mechanical analysis

## Abstract

Magnetorheological elastomers (MRE) are increasing in popularity in many applications because of their ability to change stiffness by applying a magnetic field. Instead of liquid-based 1 K and 2 K silicone, thermoplastic elastomers (TPE), based on styrene-butadiene-styrene block copolymers, have been investigated as matrix material. Three different carbonyl iron particles (CIPs) with different surface treatments were used as magneto active filler material. For the sample fabrication, the thermoplastic pressing method was used, and the MR effect under static and dynamic load was investigated. We show that for filler contents above 40 vol.-%, the linear relationship between powder content and the magnetorheological effect is no longer valid. We showed how the SiO_2_ and phosphate coating of the CIPs affects the saturation magnetization and the shear modulus of MRE composites. A combined silica phosphate coating resulted in a higher shear modulus, and therefore, the MR effect decreased, while coating with SiO_2_ only improved the MR effect. The highest performance was achieved at low deformations; a static MR effect of 73% and a dynamic MR effect of 126% were recorded. It was also shown that a lower melting viscosity of the TPE matrix helps to increase the static MR effect of anisotropic MREs, while low shear modulus is crucial for achieving high dynamic MR. The knowledge from TPE-based magnetic composites will open up new opportunities for processing such as injection molding, extrusion, and fused deposition modeling (FDM).

## 1. Introduction

Magnetorheological elastomers (MRE) are viscoelastic smart composites that show variable stiffness upon application of an external magnetic field. These compounds find applications in dampers for vibration absorption [1,2], in robotics, electronics [3,4], and force/acceleration sensors [5,6,7]. Depending on how magnetoactive particles are distributed in the matrix, two types of MRE are distinguished in literature: (1) isotropic MREs have homogeneously distributed particles within the matrix. (2) For anisotropic MREs, a magnetic field is applied while the matrix is still liquid; this causes an alignment of magnetoactive particles along the magnetic field. This alignment of the magnetoactive particles often referred to as pre-structuring, which results in a higher magnetorheological effect (MR effect). 

Typically, elastomers such as silicon rubber [8] or natural rubber [9,10] are used as matrix material because of their low stiffness and hardness. To a lesser extent, thermoplastic elastomers (TPE) such as polyurethane and styrene block copolymers were investigated [11,12]. Soft magnetic carbonyl iron particles (CIP) are most often used because of the low remanent and high saturation magnetization, as well as the high permeability [13]. Burgaz et al. [14] addressed the importance of carbonyl coating by comparing bare iron particles (BIP) to CIP and reported higher agglomeration, lower matrix-filler affinity, and increased Payne effect for the former. 

The particle size and shape of the ferromagnetic particles affect the MR behavior [15]. The smaller the particle, the greater the surface area, therefore lowering the amount of free polymer, which will result in lower flexibility, expressed in high initial shear moduli, thus lowering the MR effect. The larger the used magnetic particles; however, the greater the interparticle distance that results in a reduced dipole interaction between particles, which finally results in a lower MR effect. The oxygen released from the CIP surface into the TPE can result in a decrease in elastic properties [10]. Therefore, researchers have started to coat the carbonyl iron particles with an oxide layer.

To increase the affinity between matrix and embedded particles, several surface modifications of CIP have been reported [12,16,17]. The potential benefits of a modified CIP surface are the reduced agglomeration and higher matrix-filler affinity. The smaller interfacial thickness between matrix and filler increases flexibility while simultaneously lowering the initial shear modulus affecting the MR effect. However, P. Małecki et al. [17] reported a lower MR effect in SiO_2_-coated CIP particles that were uniformly distributed in a styrene-ethylene-butadiene-styrene (SEBS) matrix.

In practice, magnetorheological dampers have to carry a certain weight, which will result in a certain static pre-strain of the MR elastomer. Therefore, in this study, we investigate the effect of static and dynamic strain on the magnetorheological effect of anisotropic MREs based on SEBS TPE and CIP particles. Two thermoplastic elastomers with different shore hardness and carbonyl iron particles with and without surface treatment were used to investigate the MR effect under static strain conditions as well as the dynamic behavior. To avoid heating the samples during the characterization, a dynamic analyzer equipped with permanent magnets was used. Additionally, the magnetization behavior of MR soft composites with various filler content, up to 60 vol.-%, was investigated using a vibrating sample magnetometer (VSM).

## 2. Materials and Methods

Two SEBS-based materials (KRAIBURG TPE GmbH & Co. KG, Waldkraiburg, Germany) with different shore hardness were used in this study. The relevant properties are shown in Table 1.

The SEBS is a thermoplastic elastomer (TPE) that contains thermoplastic and elastomeric properties at the same time. TPEs can be divided into six subgroups (ISO 18064), and one of them is styrenic block copolymers, so-called TPS, to which SEBS (styrene-ethylene-butylene-styrene co-block polymer) belongs. SEBS is a styrenic triblock copolymer and consists of soft elastomers and hard thermoplastic blocks. The SEBS of Kraiburg is a composite based on SEBS, PP (polypropylene), fillers, and stabilizers.

Three commercial carbonyl iron particles (BASF, Ludwigshafen, Germany) were used as the soft magnetic fillers for the TPE-based MR elastomers (Table 2). (1) The HS type is a CIP without surface treatment and the mean particle size (d_50_) of 1.9µm; (2) the CC type, includes a SiO_2_ coating on the particles (d_50_ of 4.7µm); and (3) the EW-I type, where the carbonyl iron particles are coated with a SiO_2_ and phosphate-layer (d_50_ of 3.4 µm).

The magnetorheological thermoplastic elastomers were mixed in a torque rheometer (Rheomix 600, Thermofisher, Karlsruhe, Germany). For high shear mixing, the TPE material was heated to 170 °C, and the CIP filler was slowly added. Finally, the feedstocks were compounded for 30 min at 30 RPM. The naming convention for the samples consisted of the last three letters of the polymer grade (either STT or STL), followed by the volume percent of the filler and the type of used CIP (either CC, HS, or EW-I). Therefore, a sample made with the TF1 STL copolymer, loaded with 50 vol.-% of HS CIP, will be named STL50HS. After mixing, the density of the feedstock was analyzed by He-pycnometer (AccuPyc 1340, Micrometrics, Norcross, GA, USA).

Cylindrical samples with a thickness of 5 mm were warm-pressed above the melting temperature of the used TPE materials at 220 °C. A 5 kN load was applied for 8 min using a cylindrical die with a diameter of 20 mm. To introduce anisotropic orientation of the CIP particles, warm-pressed samples were placed in a magnetic field of 0.8 T generated by two permanent magnets (Q-51-51-25-N, Webcraft GmbH, Uster, Switzerland) before cooling from 220 °C to room temperature actively under flowing water. Figure 1 shows the sample processing schematically.

The magnetic properties of the MRE with various filler content and different surface treatments were measured using a vibrating sample magnetometer (VSM) from Quantum Design, USA. A 10^−2^ g sample was used for the VSM measurements. A magnetic field from 0 T to 2.2 T at ~0.1 T/min was applied. Then the magnetic field was ramped down to −2.2 T at the same rate and finally back up to 2.2 T.

An R 2000 rheometer (TA Instruments, New Castle, UK) was used to measure the complex viscosity above the melting temperature (160 °C) with a constant oscillating frequency of 10 Hz and 1% strain amplitude. MR elastomers with a 2 mm thickness were fixed between two parallel plates of 25 mm in diameter. For comparison, the complex viscosity at 170 °C will be reported in the result part for the two different SEBS elastomers filled with 30 vol.-% of carbonyl iron particles, grade HS.

For static and dynamic testing, MRE samples were assembled into a double shear testing structure made of V155 grade steel for the dynamic testing. To achieve good adhesion, the surface of the V155 grade steel was sanded before applying an adhesive paste (Araldite 2011 two-component epoxy).

An Eplexor 500 machine (NETZSCH-Gerätebau GmbH, Selb, Germany) was used for the DMA testing in shear mode. The prepared sample was fixed in the DMA as shown in Figure 2a,b, respectively.

To induce a magnetic field, disc-shaped permanent magnets with a diameter of 20 mm and thickness of 2 mm (N45, Webcraft GmbH, Uster, Switzerland) were fixed at the faces of the assembled double shear structure. Pairs of 0, 2, 4, and 6 magnets in total were used to induce a magnetic field inside the magnetorheological elastomer composite samples. The intensity and homogeneity of the induced magnetic fields were calculated using finite element analysis (FEMM). In Figure 3, the calculation for one pair of magnets and the results for a higher number of paired permanent magnets is shown. The magnetic field strength calculated for 2 and 3 pairs of magnets is shown in Appendix A.

As already mentioned, in this study, we investigated the static and dynamic magnetorheological behavior of the SEBS-based composites. The composites were cycled with a constant strain amplitude of 1% at different pre-strained levels (referred to as static tests in this report). Additionally, so-called dynamic tests were performed without any pre-strain conditions. Both tests were made with a constant frequency of 10 Hz. The MRE was calculated using:(1)$$MR_{effect}=\frac{G_{s}-G_{0}}{G_{0}}\;x\;100\%$$
where *G*_0_ represents the initial shear modulus and *G_s_* the shear modulus upon application of an external magnetic field.

Static strains were set at 0.00%, 0.20%, 0.67%, 1.09%, 1.50%, 1.93%, 2.35%, and 2.75%, whereas the dynamic strains were set at 0.04%, 0.25%, 0.45%, 0.65%, and 0.85%. The test conditions for both static and dynamic experiments are shown in Figure 4.

## 3. Results

### 3.1. Effect of CIP Concentration

To investigate the influence of CIP concentration on the MRE STL-based composites, we investigated volume fractions between 10 and 60% of HS. For the static tests, a strain amplitude of 1% and a pre-strain of 1.58% were selected. For the dynamic measurements, a strain amplitude of 0.66% was chosen. All tests were performed with an applied magnetic field of 0.54 T and a constant frequency of 10 Hz. The results of the static and dynamic tests are illustrated in Figure 5. For the static and dynamic measurements, a linear relationship between the MR effect and the CIP content was observed below 50 vol.-%. 

As is illustrated in Figure 5, above 50 vol.-%, the magnetorheological effect decreases significantly. This behavior can be explained by the critical particle volume concentration (CPVC), above which linearity approximation, as described by Jolly et al. [18] and Ginder [19], is no longer valid. The point at which there is not enough matrix present to fill all the space between the particles is known as the CPVC. To verify the CPVC point, density measurements can be used (Figure 6). Based on the mixing rule, the theoretical calculated density of a composite linearly increases when filler content increases. At the point where the theoretical and measured density shows a discrepancy, the CPVC point can be defined because air voids in the composite will lower the measured density significantly.

Density measurements of samples with a concentration above 50 vol.-% confirmed this prediction. The theoretical density of the composite with 60 vol.-% CPI was higher than the actual, measured values. Based on the discrepancy, it can be assumed that for 60 vol.-% CIP, air voids between the CIP particles are present (Figure 6). This behavior is in good agreement with the theory of critical particle volume concentration (CPVC). Filling a polymeric material above the CPVC results in a porous structure which causes a significant change in composite properties. Static and dynamic MR effects show similar results [20,21]. In Figure 6, it can be seen that the torque for the 60 vol.-% CIP is almost three times higher in comparison to 50 vol.-%. This implies that the processing behavior (e.g., viscosity) will significantly increase, and significantly higher machine power during thermoplastic shaping is required. 

### 3.2. Flow Behavior at Meting Temperature of TPE Composites

The equation of motion of spherical CIP in a liquid is inversely proportional to the viscosity of the liquid [22]. For comparison reasons, complex viscosity on two different SEBS composites with 30 vol.-% CIP was investigated with a constant oscillating frequency of 10 Hz and 1% strain amplitude. Table 3 shows the complex viscosity values above the melting point of 160 °C.

The MRE based on STT has a significantly lower complex viscosity in comparison to the one based on STL. The torque after the mixing for both matrix materials confirmed these results. Therefore, it can be expected that the orientation of the carbonyl iron particles in a magnetic field will be less restrained.

### 3.3. Magnetization Saturation of the CIPs

High saturation magnetization is a crucial factor in maximizing the MR effect, as discussed by Jolly et al. [18]. In Figure 7a, the saturation magnetization in relation to filler content is shown. The results for STL type SEBS composites with three different types of CIP filler (30 vol.-% filler content) are shown in Figure 7b.

The saturation magnetization for the composites with different filler content is shown in Figure 8. It can be observed that by extrapolation of the composite values, 2 T of pure carbonyl iron can be achieved. This value is very close to the theoretical value of iron (2.2 T).

In Figure 7b, a saturation magnetization of 0.58 T, 0.65 T, and 0.62 T for STL30HS, STL30CC, and STL30EW-I were observed, respectively. Therefore, the highest saturation magnetization was obtained for the composites based on the carbonyl iron particles with a SiO_2_ coating. This is the CC grade type of CIP used in this study.

### 3.4. Effect of the Matrix Material

To evaluate the influence of the Shore hardness of styrene-ethylene-butadiene-styrene (SEBS) elastomers on the magnetorheological effect, composites with a constant filler content (30 vol.-%) carbonyl iron particles were selected. Based on the results of the magnetic saturation, the carbonyl iron particles with surface-treatment of SiO_2_ (CC grade) were used for this investigation. 

According to G. Bossis et al. [22] and Hass [23], the time required to induce alignment of particles along an external field is proportional to the viscosity of the melted matrix. Moreover, alignment of CIP in the MRE has been proven to positively influence the MR effect [22,23]. As shown in Table 3, the viscosity of 30 vol.-% STL is much higher than 30 vol.-% STT, meaning that full alignment of particles is faster in the former composite. This is expressed in the MR effect: MRE sample based on STT showed a larger static MR effect than the one obtained from STL type, as shown in Figure 9a. However, for dynamic strain, the influence of shore hardness of the thermoplastic elastomer matrix becomes more important. Therefore, the softer STL-based composite shows a higher MR effect, as illustrated in Figure 9b.

These results indicate that for thermoplastic elastomers, a low melt flow behavior increases the static MR effect thanks to the improved orientation of CIP within the matrix. On the other hand, low matrix Shore hardness resulted in a high MR dynamic effect due to the low initial stiffness of the composite. Based on these results, further experiments were continued with softer STL type styrene-butadiene thermoplastic elastomer.

### 3.5. Effect of the CIP Surface Treatment

Małecki et al. [17] studied the magnetorheological effect of isotropic MRE based on an SEBS matrix. Coating the CIP with SiO_2_ proved to negatively impact the MR effect due to the increased affinity between particles and matrix. This reduced the mobility of CIP within the composite, which in turn hindered the buildup of CIP chains in a magnetic field which results in a lower stiffening effect of the composite.

In our experiments on anisotropic SEBS-based MRE, we observed that a SiO_2_-coating on CIP particles increased the MR effect significantly under static and dynamic strain. It can be assumed that for anisotropic MRE, mobility is less important because CIP particles are already aligned in the matrix. Figure 10a,b illustrate the effect of the surface treatment on the MR effect.

The higher static MR effect could be obtained for the STL30CC (SiO_2_-surface) MRE composite. This can be explained by the higher saturation magnetization of these particles (Figure 4).

As already mentioned, the effect of the filler-matrix affinity on the MR effect is important and has been already reported in literature [17]. Both composites STL30HS and STL30CC show low G_0_ values of 0.6 MPa, while G_0_ of STL30EW-I composite increased to 0.8 MPa. The higher shear modulus can be explained by the higher filler-matrix interfacial adhesion. These results are in agreement with the studies of Wang et al. [24]. Young’s modulus of composites is affected by the polymer-filler interfacial adhesion. When the polymer–filler interfacial adhesion is weak, the composites exhibit lower modulus. Therefore, it can be assumed that the phosphate coating on top of the silica layer results in a higher interfacial adhesion. This results in a higher G_0_. Based on Figure 10, we can conclude surface treatment of CIP of static and dynamic strain result in slightly different results, especially for the combined SiO_2_ and phosphate coating.

## 4. Conclusions

Styrene-ethylene-butylene-styrene, also known as SEBS, is an important thermoplastic soft elastomer (TPE) that behaves like rubber without undergoing vulcanization. In this study, we reported the magnetorheological effect (MR effect) of two different SEBS-based composites filled with different kinds of carbonyl iron fillers. Carbonyl iron powder without surface treatment and with SiO_2_ and SiO_2_-phosphate surface treatment were used as magneto active fillers. The main application of MR elastomers is found in the damping structures. The structure will generate shear stress on the MR elastomers, which results in a static shear strain. Based on this background, we investigated the MR effect under static and dynamic strain. The source of SEBS and carbonyl iron filler affected the static and dynamic magnetorheological performance of the magnetorheological elastomer composite (MREC) significantly.

Increasing the carbonyl iron content from 10 to 60 vol.-%, we could observe a significant drop in the MR effect above 50 vol.-% of filler. This behavior is in good agreement with the theory of critical particle volume concentration (CPVC). Filling a polymeric material above the CPVC results in a porous structure which causes a significant change in composite properties. Static and dynamic MR effects show similar results.

For anisotropic MR elastomers, changing the SEBS matrix affected the static MR effect differently. Higher static MR properties can be achieved with low complex viscosity above the melting point, and the shore hardness of the SEBS plays a minor role. With a lower complex viscosity, the resistance to generate alignment of the carbonyl iron particles is lower; thus, the chain formation under a magnetic field is faster. For the dynamic MR effect, the shore hardness of the SEBS is an important parameter, as expected.

The surface treatment of the CIPs resulted in a higher magnetization saturation of MRECs with a filler content of 30 vol.-%. The highest MR effect could be obtained with a surface treatment of SiO_2_ coating. We assume that higher static and dynamic MR effect with SiO_2_ coating is caused by two phenomena: (1) SiO_2_ coated CPI show higher saturation magnetization, which will result in higher MR-effect. (2) Higher matrix-filler affinity reduced the initial shear modulus, positively affecting the MR-effect.

## Figures and Tables

**Figure 1 polymers-13-01597-f001:**
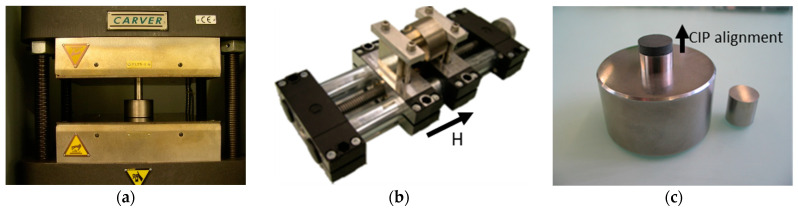
(**a**) Warm-pressing of the MRE samples, (**b**) pre-structuring the MRE sample under magnetic field H, and (**c**) CPI alignment in the TPE-based MR elastomer.

**Figure 2 polymers-13-01597-f002:**
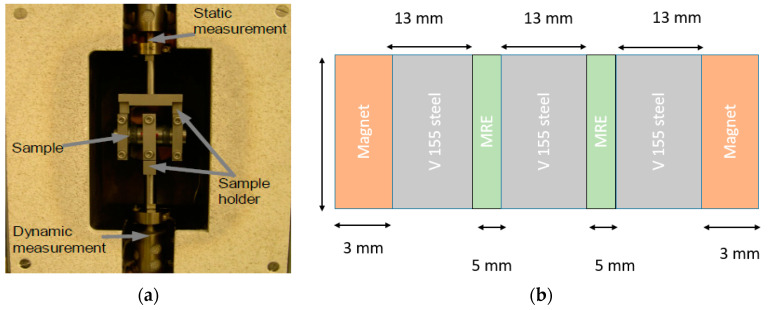
(**a**) Sample preparation in the dynamic mechanical analyzer. (**b**) A sketch of the sample with applied magnets at the edges of the sample.

**Figure 3 polymers-13-01597-f003:**
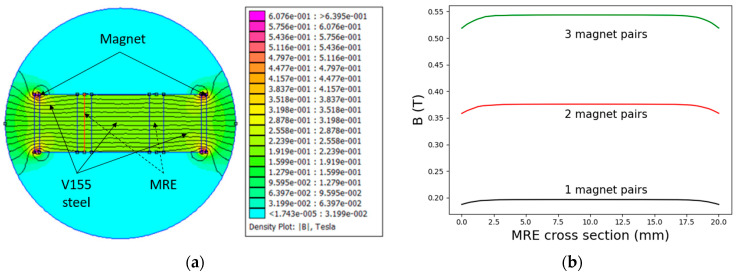
(**a**) Calculated magnetic field intensity using 1 pair of permanent magnets and (**b**) the variability of the field across the MRE.

**Figure 4 polymers-13-01597-f004:**
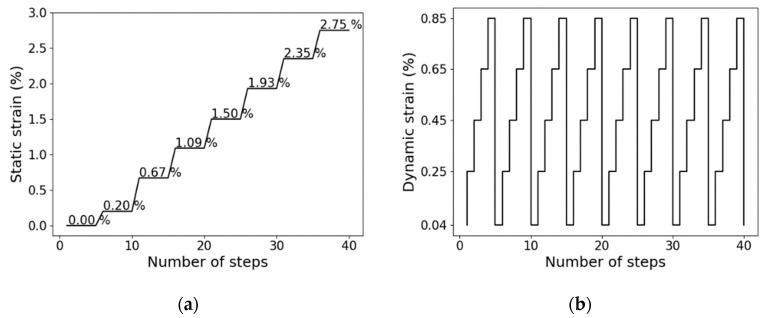
(**a**) Static and (**b**) dynamic tests procedure to investigate the MR effect of SEBS-based composites.

**Figure 5 polymers-13-01597-f005:**
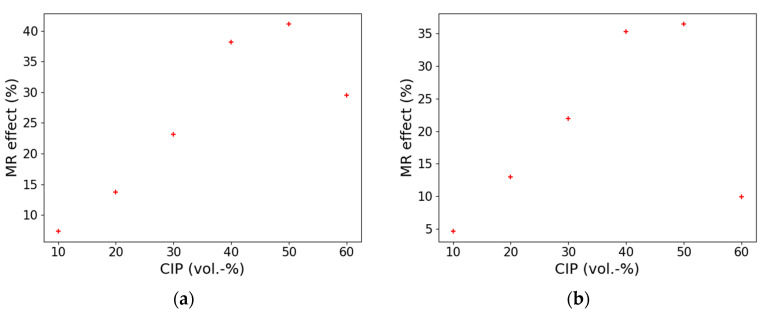
The influence of the CIP content on the MR effect (**a**) for the static test, a strain amplitude of 1% and pre-strain of 1.53% was used; (**b**) for the dynamic test, a strain amplitude of 0.66% was investigated. STL matrix with different concentrations of HS-type CIP; a constant frequency of 10 Hz and field strength of 0.54 T was selected for all tests.

**Figure 6 polymers-13-01597-f006:**
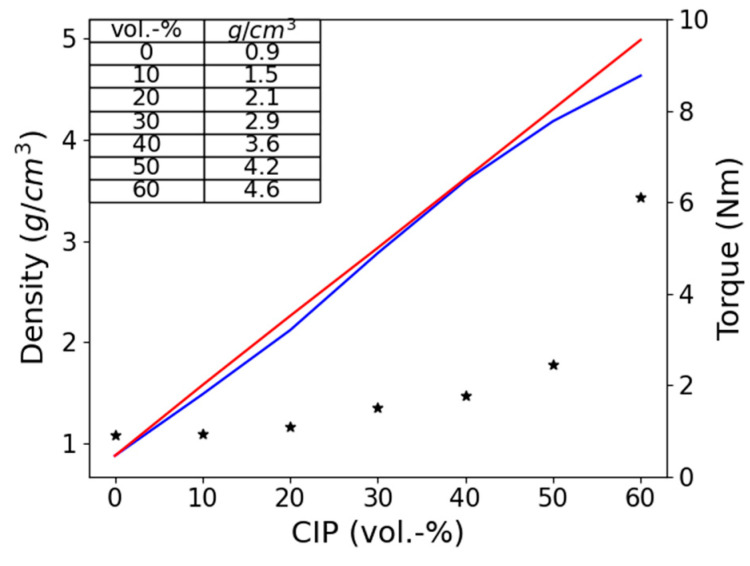
The torque at the end of the mixing process and density of the SEBS-based composite in relation to the volume concentration of CIP. The red line shows the calculated density based on the mixing rule. The blue line presents the measured density, and the stars are torque measured in Nm.

**Figure 7 polymers-13-01597-f007:**
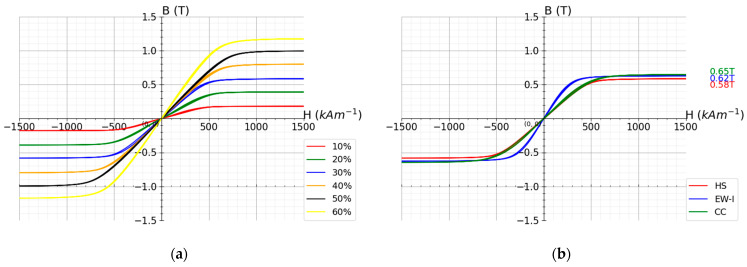
The magnetization loop for (**a**) STL type styrene-ethylene-butadiene-styrene thermoplastic elastomers with HS filler content between 10 and 60 vol.-%, (**b**) 30 vol.-% filler content of three different CIP grades in STL type styrene-ethylene-butadiene-styrene thermoplastic elastomers.

**Figure 8 polymers-13-01597-f008:**
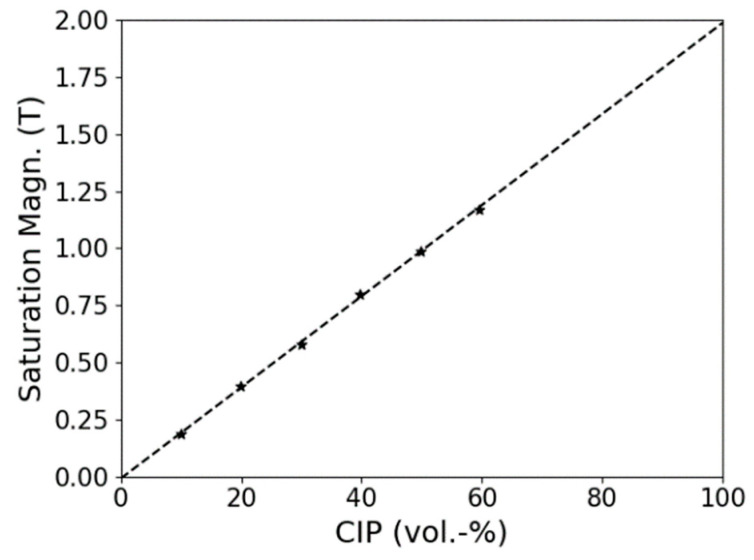
The saturation magnetization in relation to the CIP content and extrapolation of the results to 100% carbonyl iron.

**Figure 9 polymers-13-01597-f009:**
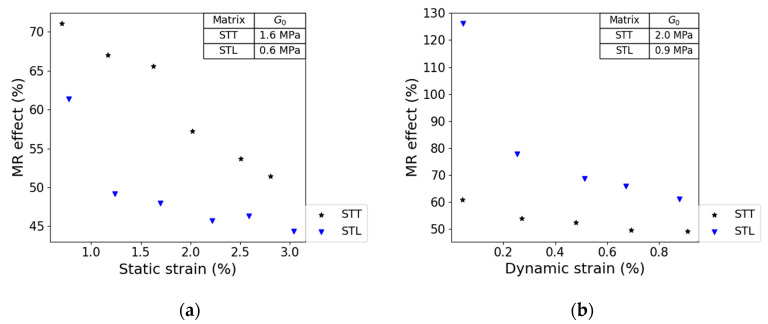
The influence of matrix material during static (**a**) and dynamic (**b**) strain on the MR effect. Composites STL30CC and STT30CC were used for this analysis. A constant strain amplitude of 1%was used for the static tests. A constant frequency of 10 Hz and field strength of 0.54 T were used for both test conditions.

**Figure 10 polymers-13-01597-f010:**
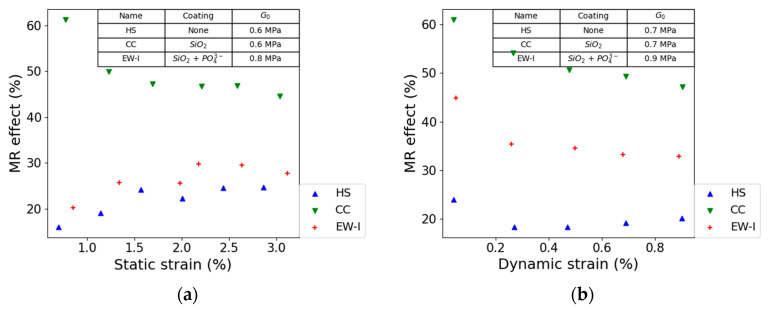
The effect of static (**a**) and dynamic (**b**) strain on MR effect. For this investigation, STL30HS, STL30CC, and STL30EW-I composites were used. All samples contain 30 vol.-% CIP in an STL matrix. A constant strain amplitude of 1%was used for the static tests. A constant frequency of 10 Hz and field strength of 0.54 T were used for both test conditions.

**Table 1 polymers-13-01597-t001:** TPE matrix materials.

	TF1 STL	TF1 STT
Density (g/cm^3^)	0.87	0.89
Hardness (Shore A)	7	15

**Table 2 polymers-13-01597-t002:** CIP materials with and without surface treatment.

Material Property	HS	CC	EW-I
Density (g/cm^3^)	7.73	7.89	7.58
d_50_ (µm)	1.9	4.7	3.4
Surface treatment	No	SiO_2_	SiO_2_ + phosphate

**Table 3 polymers-13-01597-t003:** A comparison of the complex viscosity and torque at the end of the mixing process for two different SEBS composites with 30 vol.-% CIP.

	Complex Viscosity at 170 °C (Pas)	Torque at the End of Mixing 170° (Nm)
STT30HS	184	0.1 ± 0.1
STL30HS	30,000	1.6 ± 0.2

## Data Availability

The data presented in this study are available on request from the corresponding author.

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
