# Peer review of "Analysis of Styrene-Butadiene Based Thermoplastic Magnetorheological Elastomers with Surface-Treated Iron Particles"

_polymers, 2021, doi:10.3390/polym13101597_

Round 1

Reviewer 1 Report

The aim of presented paper was to evaluate SEBS thermoplastic elastomers as matrices to anisotropic magnetorheological composites, filled with soft ferromagnetic carbonyl iron particles (CIP) with or without surface treatment. Composites based on two types of SEBS (varied in Shore hardness), containing different CIP particles ranging from 10 to 60 vol %, subjected to magnetic field during processing in order to induce an anisotropic orientation of particle within polymer matrix, were examined as regards their magnetorheological effects under static and dynamic strains, complex viscosity in melt as well as the saturation magnetization of composites in relation to the CIP type. In general the paper seems to be complex, however there are some questions arising during results analysis.

When preparing a composite, particularly with relatively high filler content, a polymer viscosity in melt is crucial for homogenous distribution of particles in polymer matrix. It’s getting even more significant if the particles are forced to motion under external magnetic field. As stated in the paper “the time required to induce alignment of particles along an external field is proportional to the viscosity of the melted matrix”. The difference in complex viscosities for composites based on two kinds of SEBS matrices, STT and STL, is huge (184 vs 30 000 Pa*s), and if so, there should be no doubt that these results are accurate. Did you observe also such a big difference for neat polymers? How many tests of complex viscosity were performed to confirm these results? What could be a reason of such a big difference in melt behaviour of polymers? In my opinion Authors should put more attention to these results and comments. On the other hand the STT matrix should be more effective in filler distribution and specified alignment of particles under magnetic field, so why the STL composites are considered to study the effect of CIP concentration in Section 3.1? The quality of the paper is high enough to recommend it to publication, however I'd expect more comments about the effect of polymer melt viscosity on MRE composites performance.

I’ve got also a few comments to the manuscript:

  1. Section 3.1., lines 152 – 154 should be removed (figure caption).
  2. 5 – kind of particles is no specified (HS, CC, EW-I?).
  3. 6 – determined density values for composites with different filler content should be added
  4. Line 197 – Figure 7b I guess
  5. Line 202 – Shore hardness
  6. Lines 221 – 224: “low melt flow behaviour and the low matrix shore hardness will result in anisotropic magnetorheological elastomers….”. Well, STT is low melt behaviour, and STL is low Shore hardness, so which is more effective? This paragraph should be modified.
  7. Lines 227 – 228: this sentence is incomprehensible.
  8. Line 233: Figures 10a and 10b?    

Author Response

We would like to thank the reviewer for the time and effort that was spent, reading our manuscript and evaluating our work. Additionally, we would like to thank the reviewer for the constructive comments and suggestions that will help improve the quality of our manuscript. The response to the comments follows:

The difference in complex viscosities for composites based on two kinds of SEBS matrices, STT and STL, is huge (184 vs 30 000 Pa*s), and if so, there should be no doubt that these results are accurate. Did you observe also such a big difference for neat polymers? How many tests of complex viscosity were performed to confirm these results? What could be a reason of such a big difference in melt behaviour of polymers?

Answer:

We agree that the difference is huge, however, we can see a huge difference in the torque after mixing too, and therefore we the that the results are correct. We added the torque results to have a second  analysis method to confirm this behavior. We also got in contact with the supplier Kraiburg and they confirmed this behavior.

On the other hand the STT matrix should be more effective in filler distribution and specified alignment of particles under magnetic field, so why the STL composites are considered to study the effect of CIP concentration in Section 3.1?

Answer:

We would like to thank the reviewer to mention this point. As shown in Figure 9a the static strain is more affected by the lower viscosity of the STT. The dynamic effect is more affected by the Shore hardness. Dampers are typically dynamically used. Therefore STL was used for further investigations. We added a part on page 8 line 227ff to make it more understandable for the reader.

Section 3.1., lines 152 – 154 should be removed (figure caption).

Answer:

Thank you for your comment. We changed this in the manuscript.

5 – kind of particles is no specified (HS, CC, EW-I?).

Answer:

Thank you for your comment. We changed this in the manuscript.

6 – determined density values for composites with different filler content should be added

Answer:

Thank you for your comment. We added this in the figure 6.

Line 197 – Figure 7b I guess

Answer:

Thank you for your comment. We changed this in the manuscript.

Line 202 – Shore hardness

Answer:

Thank you for your comment. We changed this in the manuscript.

Lines 221 – 224: “low melt flow behaviour and the low matrix shore hardness will result in anisotropic magnetorheological elastomers….”. Well, STT is low melt behaviour, and STL is low Shore hardness, so which is more effective? This paragraph should be modified.

Answer:

Thank you for your comment. We changed this in the manuscript.

Lines 227 – 228: this sentence is incomprehensible.

Answer:

Thank you for your comment. We changed this in the manuscript.

Line 233: Figures 10a and 10b?  

Answer:

Thank you for your comment. We changed this in the manuscript.

Reviewer 2 Report

The authors fabricated magnetorheological elastomers using different commercial carbonyl iron particles and thermpolastic matrices and investigated their magnetorheological properties. This is a mainstream research without particularly exciting results. I believe the paper is technically correct. Several issues have to be addressed before I can recoomen this paper for publication.

1) Page 1: Please replace "soft ferromagnetic" by soft magnetic, "low remnant" by low remanent field;

2) Page 2: "a torque rheometer" (Rheomix 600) --> Is it a rheometer, not a mixer?

3) Figures 2 and 3: Please provide a picture where the readers can see how you attached permanent magnets.

4) You show only the data for the magnetorheological effect. Please provide a table where it is possible to see the initial shear moduli of all the samples.

5) Formula 1: I believe the G should be the shear storage modulus. Please provide the data for the magnetic field dependences of the shear storage modulus, shear loss moduls and the loss tangent, both for the elastomer matrices and composites.

6) Page 5: "Based on the discrepancy, it can be assumed that
for 60 vol.-% CIP, air voids between the CIP particles are present (Figure 6)". --> The theories of Jolly and Ginder are all in the linear approximation. Obviously, they cannot describe the nonlinear effects. Above 40 vol%, your composite material is highly likely above the percolation threshold. It could be that at such high particle loadings the inclusions can not re-arrange in an magnetic field. I suggest to search for the terms "percolation theory" and "magnetoactive elastomers" for the actual works in the field.

7) Figure 8: The saturation magnetization is given in mT. It is too small. May be, the values are in Tesla?

8) You refer to the theory of the critical particle volume concentration. I am not familiar with such a theory. Please provide the corresponding references. What is the microstructure of your composite materials above the critical concentration? Are there three phases: polymer, particles and air? If yes, why is the material not degassed? The critical particle concentration refers to this three-phase composite? Could you provide micrographs confirming the existence of porosity in your materials? Are there only two phases below the critical particle concentration?

9) Commercial particles with different coating are used. These commercial particles were not developed for magnetorheological polymers. It cannot be expected that they perfectly fit for them. "The higher shear modulus can be explained by higher filler-matrix affinity" --> Which type of affinity should it be? What kind of chemical treatment is required for CIP? Please specify the affinity in detail.

10) Three types of CIP are used. However, the data on concentraition depenedence of saturation magnetization are given only for one type of particles (Figure 8). Please provide the data for all types of particles and for the corresponding powders.

11) My main objection is that paper is mostly purely descriptive. I recommend that the authors better present the objectives of their research and concentrate on the explanations of the obtained results from the physical and chemical points of view. Since this paper is submitted to a polymer journal, I would also expect more details about the chemical structure and chemical synthesis of elastomer materials involved.

Author Response

We would like to thank the reviewer for the time and effort that was spent, reading our manuscript and evaluating our work. Additionally, we would like to thank the reviewer for the constructive comments and suggestions that will help improve the quality of our manuscript. The response to the comments follows:

Page 1: Please replace "soft ferromagnetic" by soft magnetic, "low remnant" by low remanent field.

Comment:

We would like to thank the reviewer to mention this point. Different classes of magnetic materials exist. Diamagentic, paramagentetic, ferromagnetic and ferrimagnetic materials. Ferromagnetic and ferrimagnetic classes can be further divided into soft and hard ferromagnetic materials. In literature, you can find both names, soft ferromagnetic or soft magnetic materials.

Answer:

Both words were changed in the article.

Page 2: "a torque rheometer" (Rheomix 600) --> Is it a rheometer, not a mixer?

Answer:

We would like to thank you for the question of the reviewer. A torque rheometer and can be used for compounding (mixing). It is called a torque rheometer because the torque and the rotation of the rotor can be transformed into apparent viscosity and shear rate. However, there is not an exact shear rate that can be calculated. The reviewer can get more information by:

Ismael, M. R.; Clemens, F.; Bohac W. M.; Graule, T.; Hoffmann, M.J.; "Effects of rheology on the interface of Pb(Zr, Ti)O-3 monofilament composites obtained by co-extrusion," Journal of the European Ceramic Society 29 (2009), 3015-3021.

Figures 2 and 3: Please provide a picture where the readers can see how you attached permanent magnets.

Answer:

Thank you for your comment. We added some words in the figure text 2 to make it clear and we added in Figure 3 a sketch and change the figure text.

You show only the data for the magnetorheological effect. Please provide a table where it is possible to see the initial shear moduli of all the samples.

Answer:

We would like to thank for this comment, and we understand the problem the reviewer addressed to. Reading literature and writing the paper we tried to found a solution for this. As the reviewer can see, we always write the initial shear modulus in each figure. Therefore the shear modulus can be simply recalculated by the reader if needed.

Formula 1: I believe the G should be the shear storage modulus. Please provide the data for the magnetic field dependences of the shear storage modulus, shear loss modulus and the loss tangent, both for the elastomer matrices and composites.

Answer:

We would like to thank the reviewer to mention this point. We agree that the magnetorheological effect is a relative value. Because of this, we decided to add the initial shear modulus of the material without an applied magnetic field. Due to this, the MR effect can be better described. We are aware that some authors also sowed the loss modulus. However, using loss modulus for the interpretation is not useful.

Page 5: "Based on the discrepancy, it can be assumed that for 60 vol.-% CIP, air voids between the CIP particles are present (Figure 6)". --> The theories of Jolly and Ginder are all in the linear approximation. Obviously, they cannot describe the nonlinear effects. Above 40 vol%, your composite material is highly likely above the percolation threshold. It could be that at such high particle loadings the inclusions can not re-arrange in an magnetic field. I suggest to search for the terms "percolation theory" and "magnetoactive elastomers" for the actual works in the field.

Answer:

We would like to thank the reviewer for this comment. However, we are not sure if we understand it correctly. We work far above the percolation point and we do not understand how the percolation threshold will affect our statement in the text.  

Figure 8: The saturation magnetization is given in mT. It is too small. May be, the values are in Tesla?

Answer:

Thank you for your comment. We changed this in the manuscript.

You refer to the theory of the critical particle volume concentration. I am not familiar with such a theory. Please provide the corresponding references. What is the microstructure of your composite materials above the critical concentration? Are there three phases: polymer, particles and air? If yes, why is the material not degassed? The critical particle concentration refers to this three-phase composite? Could you provide micrographs confirming the existence of porosity in your materials? Are there only two phases below the critical particle concentration?

Answer:

Thank you for your comment, I integrated two references (references 24 and 25) into the text, to give the reader the possibility learn more about this. Unforunately we do not have SEM pictures that confirm the existence of porosity because of the smearing effect during sample preparation.

Commercial particles with different coating are used. These commercial particles were not developed for magnetorheological polymers. It cannot be expected that they perfectly fit for them. "The higher shear modulus can be explained by higher filler-matrix affinity" --> Which type of affinity should it be? What kind of chemical treatment is required for CIP? Please specify the affinity in detail.

Answer:

We would like to thank for the question of the reviewer. Please have a look on Table 2 in the experimental part. All information about the CIPs are included in this table. We do not understand the comment "CIPs were not developed for MR polymers. Sorry, but BASF CIPs are commonly used in the literature. This is the reason why we selected them for our study where we wanted to look on static and dynamical MR behavior of composites, which is new.

Three types of CIP are used. However, the data on concentraition depenedence of saturation magnetization are given only for one type of particles (Figure 8). Please provide the data for all types of particles and for the corresponding powders.

Answer:

We would like to thank the reviewer to mention this point. However, all three different types of CIP have been investigated and reported in figure 7b. A study on concentration for each CIP is not necessary. It is worthwhile to mention that magnetic properties in a composite are a volume effect and therefore only an offset will be observed.

My main objection is that paper is mostly purely descriptive. I recommend that the authors better present the objectives of their research and concentrate on the explanations of the obtained results from the physical and chemical points of view. Since this paper is submitted to a polymer journal, I would also expect more details about the chemical structure and chemical synthesis of elastomer materials involved.

Comment:

This is a general comment and difficult to answer. We clearly highlighted the aim of the study and maybe it is worthwhile to mention that this paper was submitted for the special issue on magnetorheological composites:

Page 1 line 21: It was also shown that a lower melting viscosity of the TPE matrix helps to increase the static MR effect of anisotropic MREs, while low shear modulus is crucial to achieve high dynamic MR.

Page 2 line 63ff: Therefore, in this study, we investigate the effect of static and dynamic strain on magnetorheological effect of anisotropic MREs based on SEBS TPE and CIP particles.

Round 2

Reviewer 2 Report

The authors did not answer a number of my comments. Please respond.

1) "Remnant" is not a proper word, in physics one speaks about the remanent magnetization or remanence.

2) The caption to modified Figure 1 is not clear. "... longitudinal ..." Please re-phrase.

3) "The higher shear modulus can be explained by higher filler-matrix affinity" --> Which type of affinity should it be? What kind of chemical treatment is required for CIP? Please specify the affinity in detail." --> Please reply.

4) "Are there three phases: polymer, particles and air? If yes, why is the material not degassed? " --> The authors agree that that they have a porous material, i.e. is it a foam? I believe this fact should be reflected in the title of the paper. Please explain why you did not remove the air bubbles. Was it desired? If yes, why?

5) However, using loss modulus for the interpretation is not useful --> This is not true. The loss modulus is useful for the evaluation of the damping properties of the developed materials. It is also field-dependent. How does the loss tangent depend on magnetic field? Please provide the information.

6) "Since this paper is submitted to a polymer journal, I would also expect more details about the chemical structure and chemical synthesis of elastomer materials involved." --> Please address this comment. I believe it is clear enough.

Author Response

Before we answer the comments and questions of the Author we would like to highlight, that we originally submitted the paper for the special issue "Magnetic Polymer Composites: Design and Applications".

Q1) "Remnant" is not a proper word, in physics one speaks about the remanent magnetization or remanence.

Our answer: We would like to thank the reviewer to see this obvious mistake.  We corrected the word and marked in in red.

Q2) The caption to modified Figure 1 is not clear. "... longitudinal ..." Please re-phrase.

Our answer: We would like for your question. we added arrow in the figure 1b and Figure 1c.

Line 108: We changed the text

Q3) "The higher shear modulus can be explained by higher filler-matrix affinity" --> Which type of affinity should it be? What kind of chemical treatment is required for CIP? Please specify the affinity in detail." --> Please reply.

Our answer: We thank for this comment. We wrote this part new:

Lin 259: " The higher shear modulus can be explained by the higher filler-matrix interface. These results are in agreement with the studies of Wang et al. [27]. Young's modulus of composites is affected by the polymer-filler interfacial adhesion. When the polymer–filler interfacial adhesion is weak, the composites exhibit lower modulus. Therefore, it can be assumed that the phosphate coating on top of the silica layer results in a higher interfacial adhesion. This results in a higher G0."

Q4) "Are there three phases: polymer, particles and air? If yes, why is the material not degassed? " --> The authors agree that that they have a porous material, i.e. is it a foam? I believe this fact should be reflected in the title of the paper. Please explain why you did not remove the air bubbles. Was it desired? If yes, why?

Our answer:We would like to thank for the comment. However, we think we have a misunderstanding here. We never mentioned that we have air voids in our samples.

Line 169ff : "The theoretical density of the composite with 60 vol.-% CPI was higher than the actual, measured values. Based on the discrepancy, it can be assumed that for 60 vol.-% CIP, air voids between the CIP particles are present (Figure 6)."

We only explain why the MR effect in Figure 5 decrease above 50 vol.%.

Q5) However, using loss modulus for the interpretation is not useful --> This is not true. The loss modulus is useful for the evaluation of the damping properties of the developed materials. It is also field-dependent. How does the loss tangent depend on magnetic field? Please provide the information.

Our answer: Thank you for your comment but we do not fully agree with your comment. Please keep in mind the aim of your study:

Line 63ff: "Therefore, in this study, we investigate the effect of static and dynamic strain on magnetorheological effect of anisotropic MREs based on SEBS TPE and CIP particles." To our knowledge, this is the first time that someone look on the static effect of MR elastomers.

We are sorry, but we do not have the data you are asking for.

Q6) "Since this paper is submitted to a polymer journal, I would also expect more details about the chemical structure and chemical synthesis of elastomer materials involved." --> Please address this comment. I believe it is clear enough.

Our answer: We would like to thank for this comment: We added some information in the experimental section.

Line 77ff: The SEBS is a thermoplastic elastomer (TPE) that contains thermoplastic and elastomeric properties at the same time. TPEs can be divided into six subgroups (ISO 18064), and one of them are styrenic block copolymers, so-called TPS, to which SEBS (styrene-ethylene-butylene-styrene co-block polymer) belongs to. SEBS is a styrenic triblock copolymer and consists of soft elastomers and hard thermoplastic blocks. The SEBS of Kraiburg is a composite based on SEBS, PP (polypropylene), fillers and stabilizers.

Round 3

Reviewer 2 Report

The authors have reacted properly to my comments.

It is a pitty that they can not provide data on the shear loss modulus and the loss tangent, because the usage of damping properties was a justification of their research.

The authors have written themselves that they suspect air voids in highly filled composites. There is a discrepancy between this statement and claiming that there were no air voids. I advice to edit the passage in the proofs, because it can be indeed misunderstood.

I think it is not proper to talk about "higher interface". Please find a better word in the proofs.

Author Response

1) The authors have written themselves that they suspect air voids in highly filled composites. There is a discrepancy between this statement and claiming that there were no air voids. I advice to edit the passage in the proofs, because it can be indeed misunderstood.
We would like to thank the reviewer for his opinion, but we clearly mentioned in the text "The theoretical density of the composite with 60 vol.-% CPI was higher than the actual, measured values. Based on the discrepancy, it can be assumed that for 60 vol.-% CIP, air voids between the CIP particles are present (Figure 6)."

However, we added now some text and hope that it will get clearer now.

Line 168: " The point at which there is not enough matrix present to fill all the space between the particles is known as the CPVC. To verify the CPVC point, density measurements can be used (Figure 6). Based on the mixing rule, the theoretical calculated density of a composite linearly increases when filler content increases. At the point where the theoretical and measured density shows a discrepancy, the CPVC point can be defined. Because air voids in the composite will lower the measured density significantly.

2) I think it is not proper to talk about "higher interface". Please find a better word in the proofs.
Thank you for your comment, this is a good point and we changed this.

Line 265: interfacial adhesion is now used. This is in line with the wording used already in scientific papers.